# Effects of Anatomical or Non-Anatomical Resection of Hepatocellular Carcinoma on Survival Outcome

**DOI:** 10.3390/jcm11051369

**Published:** 2022-03-02

**Authors:** Jae Hyun Kwon, Jung-Woo Lee, Jong Woo Lee, Young Joo Lee

**Affiliations:** 1Department of Surgery, Hallym University Sacred Heart Hospital, Hallym University College of Medicine, Anyang-si 14068, Gyeonggi-do, Korea; ponakwon@gmail.com (J.H.K.); jongwlee@hallym.or.kr (J.W.L.); 2Division of Hepato-Biliary-Pancreatic Surgery, Department of Surgery, Asan Medical Center, University of Ulsan College of Medicine, Seoul 05535, Korea; jongw.lee0212@gmail.com

**Keywords:** carcinoma, hepatocellular, hepatectomy, treatment outcome, prognosis, propensity score

## Abstract

Background: The relative benefit of anatomical resection (AR) versus non-anatomical resection (NAR) in hepatocellular carcinoma (HCC) remains controversial. This study compared the survival outcomes and recurrence rates of HCCs analysed according to tumour size and the extent of resection. Methods: Consecutive patients with HCC who underwent curative resection at Asan Medical Center between January 1999 and December 2009 were included in this study. We performed propensity score matching (PSM) according to tumour size to compare the survival outcomes between AR and NAR. A total of 986 patients were analysed; 812 and 174 patients underwent AR and NAR, respectively. Results: Before PSM, regardless of tumour size, the AR group demonstrated significantly better 5-year overall survival (OS) and recurrence-free survival (RFS) than the NAR group (*p* < 0.001). After PSM, the AR group demonstrated better OS and RFS rates than the NAR group when tumour size was less than 5 cm, but there was no significant difference in the OS and RFS rates between the two groups when tumour size was equal to or greater than 5 cm. In tumours less than 5 cm in size, AR was the most significant factor associated with OS and RFS. However, this prognostic effect of AR was not demonstrated in tumours with sizes equal to or greater than 5 cm. Conclusion: In patients with HCCs smaller than 5 cm, AR reduced the risk of tumour recurrence and improved OS. In HCCs larger than 5 cm, AR and NAR showed comparable survival outcomes.

## 1. Introduction

Primary liver cancer, of which 75–85% is constituted by hepatocellular carcinoma (HCC), is the sixth most commonly diagnosed cancer and the fourth leading cause of cancer-related death worldwide [1,2]. Hepatectomy is the standard and first-line treatment for HCC in patients with preserved hepatic reservoirs. The practice guidelines for HCC recommend resection as the treatment of choice for single or limited-numbered HCCs without significant cirrhosis or portal hypertension [3,4]; however, the guidelines are unclear about whether anatomical resection (AR) or non-anatomic resection (NAR) is superior. The extent of resection for HCC has long been under debate. 

AR involves the systematic removal of hepatic segments delineated by tumour-bearing portal tributaries [5,6] and usually involves two or more hepatic segments. In contrast, NAR involves tumour removal with a margin of uninvolved tissue. NAR is a parenchyma-sparing alternative to AR and may benefit patients with cirrhosis or poorer liver function. 

Several studies and meta-analyses have demonstrated the superiority of AR over NAR in terms of long-term survival outcome and recurrence [7,8,9,10,11]. On the other hand, while a review of well-designed comparative studies suggested comparable outcomes between AR and NAR [12,13,14,15,16,17,18,19,20,21], these studies examined solitary and primary HCCs less than 5 cm in size. Thus, this study analysed the survival outcomes of AR and NAR of HCCs including multiple lesions and different tumour sizes.

## 2. Methods

### 2.1. Study Population

Consecutive adult patients (≥18 years) who underwent hepatectomy for HCC at Asan Medical Center, Seoul, Korea, between January 1999 and December 2009 were included in this study. Patients who underwent curative resection were included in this study, whereas patients who underwent non-curative resection for palliative care or R1 and R2 resection, patients with ruptured HCC, patients who underwent re-resection for recurrent HCC, and patients with combined HCC/cholangiocarcinoma in the final pathology were excluded. Patients with vascular invasion on preoperative imaging or pathological results were all included as long as curative resection was achieved. All patients were followed up for at least five years until December 2014. The primary endpoint of this study was overall patient survival. The secondary endpoint was the recurrence of the tumour after curative resection. Patients were divided into two groups according to tumour size (smaller than and equal to or larger than 5 cm).

### 2.2. Surgical Technique and Decision Regarding the Extent of Resection

All procedures were performed by a single surgeon with more than 20 years of experience in hepatobiliary surgery and liver transplantation. Anatomical hepatectomy was the primary choice in all patients, and the extent of hepatectomy was individualised according to the estimated remnant liver volume and functional hepatic reservoir. Estimated remnant liver volume was assessed based on computed tomography volumetry, tumour location, and hepatic functional reserve. Hepatic function was assessed with the Child–Turcotte–Pugh (CTP) score and indocyanine green (ICG) retention test. Portal hypertension was evaluated based on platelet count, presence of splenomegaly, and presence of varix on endoscopic findings.

Intraoperatively, the extent of the resection in AR was determined by the Glissonean approach, which involved ligation of the tumour-locating Glissonean pedicle. Parenchymal dissection was performed using a Cavitron ultrasonic surgical aspirator (Integra LifeSciences, Plainsboro, NJ, USA) combined with the Kelly clamp-crushing technique and intermittent Pringle manoeuvre. The Pringle manoeuvre was performed for 15 min, followed by 5 min of de-clamping. AR required one Pringle clamping manoeuvre for sufficient transection of one cross-section of the liver parenchyma. For example, one Pringle manoeuvre was required for right hepatectomy, whereas two Pringle manoeuvres were required for right anterior sectionectomy.

### 2.3. Statistical Analysis

Propensity score matching was performed using SAS version 9.4 (SAS Institute, Inc., Cary, NC, USA). All other analyses were performed using R statistical software, version 3.6.3 (R Foundation for Statistical Computing, Vienna, Austria). Descriptive statistics for numerical variables were recorded as mean ± standard deviation or median with interquartile range, and categorical variables were presented as relative frequencies (percentages). We used the chi-square or Fisher’s exact test to compare categorical data and the Student’s *t*-test or Kruskal–Wallis test to compare numerical data. To address the possibility of confounding differences between AR and NAR and selection bias, propensity score matching (PSM) was performed between the AR and NAR groups with a 1:n greedy nearest-neighbour algorithm within specified calliper widths. The matching variables included baseline clinical characteristics such as age, sex, CTP class, primary liver disease, preoperative history of transarterial chemoembolisation (TACE), preoperative levels of the tumour marker alpha-fetoprotein (AFP), platelet count, total bilirubin, and ICG retention rate at 15 min, along with tumour characteristics such as number of tumours, Edmondson–Steiner grade, and macrovascular and microvascular invasion. The adequacy of the matching was described with a standardised mean difference (SMD) value; an SMD < 0.1 was considered balanced. Univariate and multivariate analyses were performed using a Cox proportional hazards regression model. Patient survival (including OS and RFS) was analysed using the Kaplan–Meier method and compared using the log-rank test. Survival comparison between the PS-matched groups was made using Cox proportional hazards regression with robust variance estimator to account for clustering by matched pairs. Differences were considered statistically significant at *p* < 0.05.

### 2.4. Ethical Considerations

The study was approved by the Institutional Review Board of Hallym University Sacred Heart Hospital, Hallym University College of Medicine, Korea (approval number: 2020-04-011). The same review board waived the requirement for informed consent because of the retrospective nature of the analyses.

## 3. Results

### 3.1. Patient Demographics and Clinical Characteristics

A total of 986 patients were included in this study, and 812 and 174 patients were categorised into the AR and NAR groups, respectively. The demographics and clinical features of the patients are described in Table 1. There were significant differences in the primary liver disease and liver function between the AR and NAR groups. The AR group demonstrated better functional liver reservoirs than the NAR group. The maximum tumour size was significantly larger in the AR group than in the NAR group. We further divided patients according to tumour size (smaller than 5 cm and equal to or larger than 5 cm) and performed PSM. The SMD values demonstrated that the matching was well balanced between the two groups (Table 2 and Table 3).

### 3.2. Survival Outcomes of the Entire Study Population

Before PSM, the data demonstrated significant differences in the 5-year OS and RFS rates between the AR and NAR groups (Figure 1A,B).

In patients with HCCs smaller than 5 cm, the 5-year OS rates in the AR and NAR groups were 79.5% and 61.8%, respectively. In contrast, in patients with HCCs equal to or larger than 5 cm, the 5-year OS rate in the AR and NAR groups were 55.7% and 47.1%, respectively.

In patients with HCCs smaller than 5 cm, the 5-year RFS rates in the AR and NAR groups were 50.2% and 36.6%, respectively. In contrast, in patients with HCCs equal to or larger than 5 cm, the 5-year RFS rates in the AR and NAR groups were 32.9% and 27.5%, respectively. Irrespective of tumour size, the AR group showed significantly better OS and RFS than the NAR group.

### 3.3. Survival Outcomes after PSM

We performed PSM and reanalysed the above data. In patients with HCCs smaller than 5 cm, the 1-, 3-, and 5-year OS rates of the NAR group were 96.6%, 84.6%, and 62.4%, respectively, whereas those of the AR group were 98.2%, 91.5%, and 78.1%, respectively. In contrast, in patients with HCCs equal to or larger than 5 cm, the 1-, 3-, and 5-year OS rates of the NAR group were 76.5%, 60.8%, and 47.1%, respectively, whereas those of the AR group were 86.5%, 58.1%, and 48.6%, respectively.

In patients with HCCs smaller than 5 cm, the 5-year OS rates of patients in the AR and NAR groups were 78.1% and 62.4%, respectively (HR = 0.53; 95% CI: 0.36–0.78; *p* = 0.001); the 5-year OS rate in the AR group was significantly better than that in the NAR group (Figure 2A). In contrast, in patients with HCCs equal to or larger than 5 cm, the 5-year OS rates in the AR and NAR groups were comparable (48.6% vs. 47.1%; HR = 0.90; 95% CI: 0.57–1.42; *p* = 0.644) (Figure 2B).

In patients with HCC smaller than 5 cm, the 1-, 3-, and 5-year RFS rates of the NAR group were 70.9%, 49.6%, and 36.8%, respectively, whereas those in the AR group were 85.3%, 58.0%, and 48.2%, respectively. In contrast, in patients with HCCs equal to or greater than 5 cm, the 1-, 3-, and 5-year RFS rates of the NAR group were 52.9%, 37.3%, and 27.5%, respectively, whereas those in the AR group were 58.8%, 39.2%, and 30.4%, respectively.

In patients with HCCs smaller than 5 cm, the 5-year RFS rates of the AR and NAR groups were 48.2% and 36.8%, respectively (HR = 0.70; 95% CI: 0.54–0.92; *p* = 0.009) (Figure 3A), with the AR group demonstrating better RFS than the NAR group. There was no significant difference in RFS between the AR and NAR groups when the tumour was equal to or greater than 5 cm (Figure 3B). In patients with HCCs equal to or greater than 5 cm, the 5-year RFS rates of the AR and NAR groups were 30.4% and 27.5%, respectively (HR = 0.87; 95% CI: 0.59–1.27; *p* = 0.472).

After PSM, the OS and RFS rates of patients with tumours smaller than 5 cm in the AR group were significantly better than those in the NAR group. For tumours equal to or larger than 5 cm, there was no significant difference in the OS and RFS between the AR and NAR groups.

### 3.4. Risk Factors for Survival Outcomes and Recurrence of HCC

Cox regression analysis demonstrated that AR affected OS outcomes in tumours smaller than 5 cm (Table 4) but did not have this effect in tumours larger than 5 cm. The risk factors for OS in tumours smaller than 5 cm included increased aspartate aminotransferase (AST) (log-transformed) and low albumin levels, multiple tumours, and high Edmondson–Steiner grades. In tumours equal to or larger than 5 cm, only multiple tumours were a significant risk factor for OS.

Cox regression analysis also demonstrated that AR was associated with improved RFS in tumours less than 5 cm in size (Table 5) but not in tumours equal to or greater than 5 cm in size. The risk factors for RFS in tumours smaller than 5 cm included male sex, increased AST levels, low albumin levels, and multiple tumours. In tumours equal to or greater than 5 cm in size, male sex, hepatitis C virus-associated primary liver disease, platelet count, AST level, prothrombin time, and multiple tumours were associated with RFS.

## 4. Discussion

This retrospective study included consecutive patients who underwent curative hepatectomy for HCC divided into PS-matched cohorts based on the extent of resection. Our data demonstrated that the extent of hepatic resection (AR or NAR) had different impacts on the survival outcome and recurrence rate of HCC when HCC was analysed according to tumour size. For tumours smaller than 5 cm, AR demonstrated superior survival outcomes (both OS and RFS) compared to NAR. Conversely, for tumours equal to or larger than 5 cm, the survival outcomes of AR and NAR were comparable.

AR is based on the high propensity of HCC to invade intrahepatic vasculature by spreading through the closest portal veins [22]. Systematic removal of tumour-bearing portal territories may eliminate potential micro-metastases near the tumour [22], which theoretically improves survival outcomes by reducing the risk of recurrence. However, the extent of hepatic resection should consider the size and location of the tumour, as well as the hepatic functional reservoir. Patients have different hepatic functional reservoirs and baseline clinical characteristics, which make it difficult to predict the prognosis of AR and NAR. PSM overcomes some of these biases. Some real-world studies have demonstrated the superior overall or disease-free survival benefit of AR over NAR [7,8,9,10,11], whereas other studies have found no such benefits [12,13,14,15,16,17,18,19,20,21].

AR is known to be associated with longer operating times and more intraoperative bleeding compared to NAR. Through technical innovations in surgery, improvements in perioperative care, and enhanced understanding of liver anatomy, morbidity and mortality after hepatic resection are now acceptable and conquerable. Despite these advances, post-hepatectomy liver failure (PHLF) still remains an unresolved and devastating morbidity after major hepatectomy [23]. More functional liver parenchyma is removed in AR, which makes patients undergoing AR more vulnerable to morbidity and mortality, particularly from PHLF, than those undergoing NAR. When determining the extent of resection, hepatobiliary surgeons should weigh the risk of PHLF and tumour recurrence. The operating times for AR in this study were significantly longer than for NAR; however, the duration of postoperative hospital stay and morbidity rates were comparable between the AR and NAR groups (Table 1).

The criteria for patient selection and the extent of hepatectomy tend to differ among surgeons and centres. As such, multicentre studies and studies that analyse cases performed by several surgeons at a single centre may be more prone to bias. The current study was unique because it examined consecutive cases of hepatectomy for HCC performed by a single surgeon at a single centre. To the best of our knowledge, this is the first study to analyse the outcomes of AR and NAR for HCC performed by a single surgeon. While a single-centre, single-surgeon study may be prone to selection bias, it provides consistent data, particularly in terms of the surgical technique and decision process for the extent of resection (AR vs. NAR). Further, we analysed a large number of hepatectomies performed by a single surgeon, which enhanced the quality of our study results.

The best opportunity for cure in patients with HCC is during the first instance of surgical intervention, so determining the extent and timing of resection is crucial for improving patient survival. The primary concern in HCC is recurrence following treatment. In this study, multiple HCC tumours larger than 5 cm affected overall survival outcomes, whereas for tumours smaller than 5 cm, AST and albumin levels, histologic characteristics, and the extent of resection did affect the survival outcome. Further, recurrence was more likely among male patients and those with poor liver function and multiple tumours, regardless of tumour size. AR seemed to reduce the risk of recurrence and improved survival in patients with tumours smaller than 5 cm. Our data also suggested that in tumours greater than 5 cm, the tumour biology itself may be more important than the extent of resection. Because the prognostic factors in pathology could be acquired only after the surgery (i.e., liver resection or transplantation), predicting these risk factors preoperatively through an imaging study is crucially important in deciding the best treatment choice for the patient. A recent well-designed clinical study proved a strong association between Liver Imaging Reporting and Data System (LI-RADS) classification and pathological findings, which proved to be significant prognostic factors (i.e., microvascular invasion and satellites) for patient survival outcomes [24]. Further well-designed prospective studies are needed to determine the impact of tumour biology on surgical decisions and to develop predictors for worse prognostic pathological findings in preoperative imaging studies. While our study compared AR and NAR, our data suggest that in selected patients, prolonged survival in HCC may not be achieved through AR alone.

The limitations of this study include its retrospective single-centre design and a discrepancy in the sample size between the AR and NAR groups. A selection bias, therefore, remained between the two groups even though PSM was conducted to adjust for baseline clinical characteristics and liver functions. Further randomised controlled trials are required to confirm the role of AR and NAR in HCC patients according to tumour size.

## 5. Conclusions

This retrospective PS-matched study compared the benefits of AR and NAR performed by a single surgeon for curative resection of HCC. In HCCs smaller than 5 cm, AR demonstrated a favourable survival benefit over NAR. However, for HCCs larger than 5 cm, AR and NAR demonstrated comparable survival outcomes. Further well-designed comparative studies are needed to identify which patients benefit from AR in HCC when factors other than tumour size are considered.

## Figures and Tables

**Figure 1 jcm-11-01369-f001:**
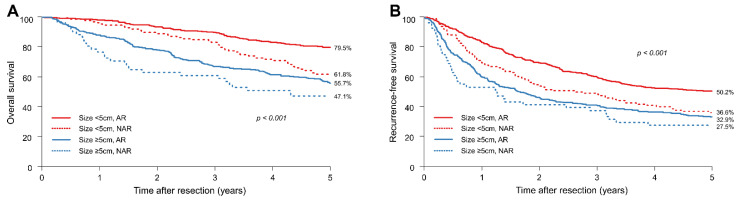
Survival outcomes of all patients. (**A**) Overall survival (OS) and (**B**) recurrence-free survival (RFS) of all patients.

**Figure 2 jcm-11-01369-f002:**
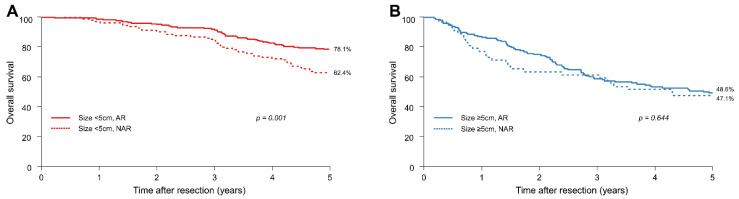
After propensity score matching, (**A**) overall survival (OS) of patients with HCCs smaller than 5 cm and (**B**) OS of patients with HCCs equal to or greater than 5 cm.

**Figure 3 jcm-11-01369-f003:**
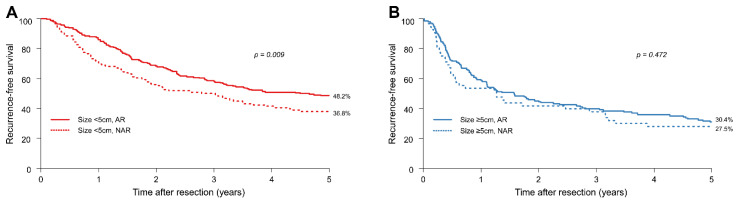
After propensity score matching, (**A**) recurrence-free survival (RFS) of patients with HCCs smaller than 5 cm and (**B**) RFS of patients with HCCs equal to or greater than 5 cm.

**Table 1 jcm-11-01369-t001:** Demographics and clinical characteristics of patients.

	NAR (*n* = 174)	AR (*n* = 812)	*p*-Value
Age (years)	54 (49–61)	54 (47–61)	0.296
Sex			0.081
Male	145 (83.3)	628 (77.3)	
Female	29 (16.7)	184 (22.7)	
Child–Turcotte–Pugh class			0.145
A	172 (98.9)	810 (99.8)	
B	2 (1.1)	2 (0.2)	
Primary liver disease			0.016
HBV	136 (78.2)	642 (79.1)	
HCV	17 (9.8)	38 (4.7)	
NBNC	21 (12.1)	132 (16.3)	
ASA PS classification			0.100
Ⅰ	21 (12.1)	114 (14.0)	
Ⅱ	127 (73.0)	616 (75.9)	
Ⅲ	26 (14.9)	82 (10.1)	
Preoperative TACE	44 (25.3)	169 (20.8)	0.193
Preoperative laboratory result			
AFP (ng/mL)	30.0 (8.0–484.0)	50.0 (6.0–657.0)	0.591
Platelet (×10^3^/uL)	135.50 (100.0–177.8)	152.00 (119.0–195.3)	<0.001
AST (IU/L)	36.00 (29.0–50.8)	36.00 (27.0–52.0)	0.897
ALT (IU/L)	32.00 (22.0–43.0)	33.00 (22.0–51.0)	0.229
Total bilirubin (mg/dL)	0.90 (0.80–1.20)	0.90 (0.70–1.10)	0.008
Albumin (g/dL)	3.70 (3.5–4.0)	3.80 (3.5–4.1)	0.019
PT (%)	87.45 (78.2–95.9)	90.40 (83.0–98.9)	<0.001
PT (INR)	1.08 (1.03–1.16)	1.06 (1.01–1.11)	<0.001
ICG R15	13.12 (8.5–19.1)	11.38 (7.8–15.9)	0.004
Operation time (min)	171.00 (135.0–215.0)	180.00 (150.0–225.3)	0.003
Postoperative hospital stay (days)	13.00 (12.0–16.0)	14.00 (12.0–17.0)	0.306
Postoperative complication	21 (12.1)	99 (12.2)	0.964
Pathology			
Maximum tumour size (cm)	3.5 (2.4–5.0)	4.2 (2.8–7.0)	<0.001
Number of tumours			0.215
Solitary	147 (84.5)	714 (87.9)	
Multiple	27 (15.5)	98 (12.1)	
Edmondson–Steiner grade			0.093
I–II	78 (44.8)	421 (51.8)	
III–IV	96 (55.2)	391 (48.2)	
Macrovascular invasion	19 (10.9)	114 (14.0)	0.274
Microvascular invasion	25 (14.4)	150 (18.5)	0.198

AFP, alpha-fetoprotein; ALT, alanine aminotransferase; ASA PS, American Society of Anesthesiologists physical status; AST, aspartate aminotransferase; AR, anatomic resection; HBV, hepatitis B virus; HCV, hepatitis C virus; ICG R15, indocyanine green retention rate at 15 min; INR, international normalised ratio; NAR, non-anatomic resection; NBNC, non-HBV non-hepatitis C virus; PT, prothrombin time; TACE, transcatheter arterial chemoembolisation.

**Table 2 jcm-11-01369-t002:** Baseline characteristics of patients before and after propensity score matching in tumour sizes less than 5 cm.

	Total Set				Matched Set *		
Tumour Size < 5 cm	NAR Group	AR Group	*p*-Value	SMD	NAR Group	AR Group	SMD
	*n* = 123	*n* = 478			*n* = 117	*n* = 224	
Age							
<50 years	31 (25.2)	149 (31.2)	0.433	0.133	31 (26.5)	68 (30.4)	0.100
50–60 years	59 (48.0)	209 (43.7)			56 (47.9)	97 (43.3)	
>60 years	33 (26.8)	120 (25.1)			30 (25.6)	59 (26.3)	
Sex (male/female)	103 (83.7%)/20	368 (77.0%)/110	0.105	0.171	97 (82.9%)/20	180 (80.4%)/44	0.066
Primary liver disease							
HBV	92 (74.8)	395 (82.6)	0.032	0.241	89 (76.1)	175 (78.1)	0.051
HCV	15 (12.2)	27 (5.6)			13 (11.1)	22 (9.8)	
NBNC	16 (13.0)	56 (11.7)			15 (12.8)	27 (12.1)	
Preoperative TACE	27 (22.0)	97 (20.3)	0.685	0.041	24 (20.5)	55 (24.6)	0.097
AFP (ng/mL)							
<100	80 (65.0)	294 (61.5)	0.771	0.073	76 (65.0)	139 (62.1)	0.062
100–1000	27 (22.0)	116 (24.3)			25 (21.4)	51 (22.8)	
>1000	16 (13.0)	68 (14.2)			16 (13.7)	34 (15.2)	
Platelet							
<150	77 (62.6)	290 (60.7)	0.695	0.040	73 (62.4)	131 (58.5)	0.080
≥150	46 (37.4)	188 (39.3)			44 (37.6)	93 (41.5)	
Log-transformed AST	3.62 ± 0.45	3.59 ± 0.42	0.498	0.067	3.62 ± 0.45	3.61 ± 0.43	0.025
Total bilirubin							
<1.5	110 (89.4)	444 (92.9)	0.203	0.122	106 (90.6)	207 (92.4)	0.065
≥1.5	13 (10.6)	34 (7.1)			11 (9.4)	17 (7.6)	
Albumin	3.70 ± 0.41	3.80 ± 0.39	0.008	0.265	3.73 ± 0.38	3.76 ± 0.38	0.073
PT (INR)	1.10 ± 0.10	1.28 ± 4.43	0.669	0.055	1.10 ± 0.09	1.09 ± 0.09	0.057
ICG R15							
<15	74 (60.2)	341 (71.3)	0.017	0.237	74 (63.2)	144 (64.3)	0.022
≥15	49 (39.8)	137 (28.7)			43 (36.8)	80 (35.7)	
Number of tumours							
Solitary	107 (87.0)	428 (89.5)	0.420	0.079	104 (88.9)	191 (85.3)	0.108
Multiple	16 (13.0)	50 (10.5)			13 (11.1)	33 (14.7)	
Macrovascular invasion	9 (7.3)	34 (7.1)	0.938	0.008	8 (6.8)	13 (5.8)	0.043
Microvascular invasion	11 (8.9)	47 (9.8)	0.766	0.031	10 (8.5)	20 (8.9)	0.014
Edmondson–Steiner grade							
I–II	60 (48.8)	273 (57.1)	0.097	0.168	58 (49.6)	112 (50.0)	0.009
III–IV	63 (51.2)	205 (42.9)			59 (50.4)	112 (50.0)	

AFP, alpha-fetoprotein; AST, aspartate aminotransferase; AR, anatomic resection; HBV, hepatitis B virus; HCV, hepatitis C virus; ICG R15, indocyanine green retention rate at 15 min; INR, international normalised ratio; NAR, non-anatomic resection; NBNC, non-HBV non-hepatitis C virus; PT, prothrombin time; SMD, standardised mean difference; TACE, transcatheter arterial chemoembolisation. * Propensity score matching was used with 1:n greedy nearest-neighbour algorithm within specified calliper widths. SMD < 0.1 is considered to be balanced.

**Table 3 jcm-11-01369-t003:** Baseline characteristics of patients before and after propensity score matching in tumour sizes greater than or equal to 5 cm.

	Total Set				Matched Set *		
Tumour Size ≥ 5 cm	NAR Group	AR Group	*p*-Value	SMD	NAR Group	AR Group	SMD
	*n* = 51	*n* = 334			*n* = 51	*n* = 148	
Age							
<50 years	16 (31.4)	117 (35.0)	0.711	0.123	16 (31.4)	46 (31.1)	0.033
50–60 years	22 (43.1)	124 (37.1)			22 (43.1)	66 (44.6)	
>60 years	13 (25.5)	93 (27.8)			13 (25.5)	36 (24.3)	
Sex (male/female)	42 (82.4%)/9	260 (77.8%)/74	0.466	0.113	42 (82.4%)/9	119 (80.4%)/29	0.050
Primary liver disease							
HBV	44 (86.3)	247 (74.0)	0.107	0.356	44 (86.3)	127 (85.8)	0.041
HCV	2 (3.9)	11 (3.3)			2 (3.9)	7 (4.7)	
NBNC	5 (9.8)	76 (22.8)			5 (9.8)	14 (9.5)	
Preoperative TACE	17 (33.3)	72 (21.6)	0.063	0.266	17 (33.3)	50 (33.8)	0.010
AFP (ng/mL)							
<100	28 (54.9)	163 (48.8)	0.054	0.384	28 (54.9)	80 (54.1)	0.070
100–1000	14 (27.5)	60 (18.0)			14 (27.5)	38 (25.7)	
>1000	9 (17.6)	111 (33.2)			9 (17.6)	30 (20.3)	
Platelet							
<150	27 (52.9)	109 (32.6)	0.005	0.419	27 (52.9)	71 (48.0)	0.099
≥150	24 (47.1)	225 (67.4)			24 (47.1)	77 (52.0)	
Log-transformed AST	3.77 (0.50)	3.77 (0.57)	0.988	0.002	3.77 (0.50)	3.80 (0.59)	0.055
Total bilirubin							
<1.5	49 (96.1)	311 (93.1)	0.424	0.131	49 (96.1)	140 (94.6)	0.070
≥1.5	2 (3.9)	23 (6.9)			2 (3.9)	8 (5.4)	
Albumin	3.67 (0.45)	3.74 (0.44)	0.344	0.140	3.67 (0.45)	3.66 (0.46)	0.042
PT (INR)	1.08 (0.09)	1.06 (0.09)	0.052	0.286	1.08 (0.09)	1.08 (0.09)	0.022
ICG R15							
<15	32 (62.7)	235 (70.4)	0.272	0.162	32 (62.7)	90 (60.8)	0.040
≥15	19 (37.3)	99 (29.6)			19 (37.3)	58 (39.2)	
Number of tumours							
Solitary	40 (78.4)	286 (85.6)	0.184	0.188	40 (78.4)	121 (81.8)	0.083
Multiple	11 (21.6)	48 (14.4)			11 (21.6)	27 (18.2)	
Macrovascular invasion	10 (19.6)	80 (24.0)	0.495	0.105	10 (19.6)	28 (18.9)	0.017
Microvascular invasion	14 (27.5)	103 (30.8)	0.624	0.075	14 (27.5)	44 (29.7)	0.050
Edmondson–Steiner grade							
I–II	18 (35.3)	148 (44.3)	0.226	0.185	18 (35.3)	51 (34.5)	0.018
III–IV	33 (64.7)	186 (55.7)			33 (64.7)	97 (65.5)	

AFP, alpha-fetoprotein; AST, aspartate aminotransferase; AR, anatomic resection; HBV, hepatitis B virus; HCV, hepatitis C virus; ICG R15, indocyanine green retention rate at 15 min; INR, international normalised ratio; NAR, non-anatomic resection; NBNC, non-HBV non-hepatitis C virus; PT, prothrombin time; SMD, standardised mean difference; TACE, transcatheter arterial chemoembolisation. * Propensity score matching was used with 1:n greedy nearest-neighbour algorithm within specified calliper widths. SMD < 0.1 is considered to be balanced.

**Table 4 jcm-11-01369-t004:** Multivariate Cox regression analyses for patient overall survival.

		Tumour Size < 5 cm		Tumour Size ≥ 5 cm	
		Multivariate Cox Analysis		Multivariate Cox Analysis	
		HR (95% CI)	*p*-Value	HR (95% CI)	*p*-Value
AR		0.54 (0.37–0.77)	0.001	0.84 (0.53–1.32)	0.439
Age (years)	50–60 years	0.88 (0.57–1.34)	0.540	0.73 (0.49–1.08)	0.115
	>60 years	1.28 (0.79–2.09)	0.319	0.84 (0.54–1.29)	0.421
Sex	Female	0.77 (0.49–1.21)	0.251	0.78 (0.52–1.16)	0.219
Primary disease	HCV	1.35 (0.74–2.45)	0.322	1.59 (0.75–3.37)	0.227
	NBNC	0.91 (0.51–1.62)	0.746	0.90 (0.58–1.41)	0.642
Preoperative TACE	Yes	1.43 (0.96–2.11)	0.076	1.20 (0.83–1.73)	0.332
AFP (ng/mL)	100–1000	1.36 (0.90–2.06)	0.147	0.96 (0.62–1.50)	0.856
	>1000	1.39 (0.86–2.24)	0.180	1.07 (0.71–1.61)	0.735
Platelet	≥150	0.78 (0.53–1.14)	0.190	0.74 (0.54–1.02)	0.07
Log-transformed AST		1.88 (1.29–2.76)	0.001	1.24 (0.90–1.69)	0.185
Total bilirubin	≥1.5	1.29 (0.73–2.30)	0.382	1.73 (0.99–3.01)	0.053
Albumin		0.62 (0.39–0.99)	0.044	0.81 (0.55–1.19)	0.276
PT (INR)		0.96 (0.80–1.15)	0.636	0.61 (0.08–4.70)	0.632
ICG R15	≥15	1.06 (0.72–1.56)	0.774	1.20 (0.83–1.72)	0.337
Number of tumours	Multiple	2.23 (1.46–3.41)	<0.001	2.25 (1.54–3.28)	<0.001
Macrovascular invasion		2.00 (0.72–5.60)	0.186	1.13 (0.61–2.07)	0.7
Microvascular invasion		1.61 (0.62–4.15)	0.327	1.77 (0.98–3.20)	0.058
Edmondson–Steiner grade	III–IV	1.57 (1.11–2.21)	0.011	1.26 (0.88–1.79)	0.203

AFP, alpha-fetoprotein; AST, aspartate aminotransferase; AR, anatomic resection; CI, confidence interval; HCV, hepatitis C virus; HR, hazard ratio; ICG R15, indocyanine green retention rate at 15 min; INR, international normalised ratio; NBNC, non-HBV non-hepatitis C virus; PT, prothrombin time; TACE, transcatheter arterial chemoembolisation.

**Table 5 jcm-11-01369-t005:** Multivariate Cox regression analyses for patient recurrence-free survival.

		Tumour Size < 5 cm		Tumour Size ≥ 5 cm	
		Multivariate Cox Analysis		Multivariate Cox Analysis	
		HR (95% CI)	*p*-Value	HR (95% CI)	*p*-Value
AR		0.69 (0.53–0.90)	0.006	0.93 (0.64–1.35)	0.707
Age (years)	50–60 years	0.86 (0.65–1.15)	0.312	0.77 (0.55–1.06)	0.106
	>60 years	1.26 (0.90–1.75)	0.173	0.89 (0.62–1.27)	0.518
Sex	Female	0.53 (0.38–0.73)	<0.001	0.69 (0.50–0.96)	0.030
Primary disease	HCV	1.04 (0.67–1.61)	0.857	2.07 (1.14–3.76)	0.016
	NBNC	0.73 (0.49–1.10)	0.135	0.74 (0.52–1.06)	0.104
Preoperative TACE	Yes	1.29 (0.98–1.70)	0.068	1.14 (0.84–1.55)	0.384
AFP (ng/mL)	100–1000	1.25 (0.95–1.65)	0.110	0.91 (0.62–1.32)	0.608
	>1000	0.98 (0.69–1.40)	0.915	1.14 (0.82–1.58)	0.426
Platelet	≥150	0.80 (0.62–1.03)	0.078	0.66 (0.51–0.86)	0.002
Log-transformed AST		1.48 (1.13–1.94)	0.005	1.48 (1.14–1.91)	0.003
Total bilirubin	≥1.5	0.94 (0.62–1.42)	0.778	1.50 (0.91–2.49)	0.112
Albumin		0.56 (0.41–0.77)	<0.001	0.82 (0.59–1.14)	0.227
PT (INR)		0.96 (0.84–1.09)	0.503	0.13 (0.02–0.71)	0.019
ICG R15	≥15	1.00 (0.77–1.29)	0.981	1.22 (0.89–1.66)	0.214
Number of tumours	Multiple	1.48 (1.08–2.04)	0.015	2.20 (1.58–3.06)	<0.001
Macrovascular invasion		1.53 (0.71–3.30)	0.282	1.24 (0.71–2.16)	0.448
Microvascular invasion		1.40 (0.70–2.82)	0.340	1.17 (0.69–1.99)	0.554
Edmondson–Steiner grade	III–IV	1.21 (0.96–1.52)	0.112	1.06 (0.80–1.41)	0.661

AFP, alpha-fetoprotein; AST, aspartate aminotransferase; AR, anatomic resection; CI, confidence interval; HCV, hepatitis C virus; HR, hazard ratio; ICG R15, indocyanine green retention rate at 15 min; INR, international normalised ratio; NBNC, non-HBV non-hepatitis C virus; PT, prothrombin time; TACE, transcatheter arterial chemoembolisation.

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
