# Peer review of "Effects of Anatomical or Non-Anatomical Resection of Hepatocellular Carcinoma on Survival Outcome"

_jcm, 2022, doi:10.3390/jcm11051369_

Round 1

Reviewer 1 Report

Thank you for the opportunity to revise this interesting paper.
There could be some biases by following issues:

  • the limitations of the study need to be further discussed;
  • Were patients with preoperative vascular infiltration resected or defined as non-candidates for surgery? In the histological report (tables 2 and 3) there are patients with macrovascular invasion, the data could lead to confusion.
  • Treatment of HCC has varied in part over the years, as have nonsurgical therapies. In the period of the study, from 1999 to 2009, which guidelines were followed to treat HCC? Why haven't the Authors analyzed more recent and updated data?
  • 70/71 line typo in pt of the text. 

Author Response

We are grateful for the opportunity to revise our manuscript and the helpful comments of the reviewers. We have revised our manuscript accordingly. 

Reviewer 2 Report

In their original paper entitled "Effects of anatomical or non-anatomical resection of hepatocellular carcinoma on survival outcome" Jae Hyun Kwon and coll. provides a retrospective single-institution analysis focusing on the results of surgical resection for HCC, analysing the impact of the extension of the hepatectomy on survival outcomes.

The best approach to resectable HCC is still debated: several Authors support anatomic resection (AR) in order to enhance the oncological outcomes by removing the site of possible satellites micronodules that could affect an early recurrence.
On the other hand, extended resection could increase the risk of post-hepatectomy liver failure, and thus the application of limited non-anatomic resection (NAR) is strongly supported especially in less compensated patients, taking into account some evidence suggesting a non-inferiority of NAR in terms or recurrence-free and overall survival

The study population is large and the follow-up is long enough to provide a robust evidence.

The methodology is rigorous and the study limitation well acknowledged

I have some points that I wish to be addressed:

1) Please report the patient peformance status

2) Please specify if the postoperative complications that are reported accounts for only Clavien-Dindo grade >=3 

3) Please report postoperative/90-days mortality

3) Iterative treatments for HCC management are increasingly used, with important effect on patient survival; I would like to know the rate of re-resection after tumor recurrence in both patient groups (AR vs. NAR)

4) The salvage liver transplantation (SLT) is the most effective treatment for tumor recurrence; in order to role out any influence of SLT on oncological outcomes of the two patient cohorts, the Authors should show how many patients were transplanted ab-initio (ie: discovery of unfavorable pathological findings on resected specimen) or after tumor recurrence

5) Please better discuss the impact of tumor pathology on post-resection outcomes, referring to a recently published western series: 10.3390/diagnostics12010160

6) Considering the significant impact of liver function highlighted in COX regression analysis, I would suggest to perform a competing-risk analysis focusing on cancer-related death

Congratulations for the interesting paper

Best regards

Author Response

We would like to thank you for your thoughtful review of our manuscript and the invaluable comments. 

Round 2

Reviewer 2 Report

The Authors properly replied to the previous comments.